# *TaNBR1*, a Novel Wheat NBR1-like Domain Gene Negatively Regulates Drought Stress Tolerance in Transgenic *Arabidopsis*

**DOI:** 10.3390/ijms23094519

**Published:** 2022-04-20

**Authors:** Liuping Chen, Qian Lv, Weibing Yang, Hui Yang, Qiaoyan Chen, Bingxin Wang, Yanhong Lei, Yanzhou Xie

**Affiliations:** 1State Key Laboratory of Crop Stress Biology in Arid Areas, College of Agronomy, Northwest A&F University, Xianyang 712100, China; chenliupinghist@163.com (L.C.); lvqianhist@163.com (Q.L.); jy_wheat@nwafu.edu.cn (W.Y.); wangbx@nwafu.edu.cn (B.W.); 2021060044@nwafu.edu.cn (Y.L.); 2Nanyang Academy of Agricultural Sciences, Nanyang 473000, China; 13721807607@163.com; 3Henan Institute of Science and Technology, College of Life and Science, Xinxiang 453003, China; cqy20100216@163.com

**Keywords:** drought stress, negative regulation, *TaNBR1*, *Triticum aestivum*, NBR1 protein

## Abstract

Drought stress is an important factor that severely affects crop yield and quality. Autophagy has a crucial role in the responses to abiotic stresses. In this study, we explore *TaNBR1* in response to drought stress. Expression of the *TaNBR1* gene was strongly induced by NaCl, PEG, and abscisic acid treatments. The TaNBR1 protein is localized in the Golgi apparatus and autophagosome. Transgenic *Arabidopsis* plants overexpressing *TaNBR1* exhibited reduced drought tolerance. When subjected to drought stress, compared to the wild-type (WT) lines, the transgenic overexpressing *TaNBR1* plants had a lower seed germination rate, relative water content, proline content, and reduced accumulation of antioxidant enzymes, i.e., superoxide dismutase, peroxidase, and catalase, as well as higher chlorophyll losses, malondialdehyde contents, and water loss. The transgenic plants overexpressing *TaNBR1* produced much shorter roots in response to mannitol stress, in comparison to the WT plants, and they exhibited greater sensitivity to abscisic acid treatment. The expression levels of the genes related to stress in the transgenic plants were affected in response to drought stress. Our results indicate that *TaNBR1* negatively regulates drought stress responses by affecting the expression of stress-related genes in *Arabidopsis*.

## 1. Introduction

Drought stress is one of the main adverse factors for greatly decreasing the quality and productivity of grain crops, which threatens human food security.The most cost-effective and environmentally friendly way to combat drought is to breed drought-resistant varieties, and the discovery of resistance genes is crucial to the development of new drought-resistant varieties of wheat. In recent years, numerous key genes related to drought stress have been identified in several plant species [1,2,3,4,5,6,7].

In the eukaryotic organisms, the autophagy–lysosome pathway (ALP) is essential for the regulation of protein degradation in the ubiquitin–proteasome system (UPS), which controls the degradation and recycling of unwanted cells in the lysosome/vacuole [8,9]. In the ALP, p62/SQSTM1 and NBR1 act as specific receptors, which can recruit the ubiquitinated substrates to the autophagosome by interacting with the LC3/ATG8 protein, and then the ubiquitinated substrates are degraded in the autophagosome. In mammals, several autophagy receptors, including p62/sequestosome-1 (p62/SQSTM-1), NBR1, OPTN, and NDP52/CALCOCO2, have been identified [10,11,12]. P62, a widely studied autophagic cargo receptor, has several conserved structural domains, including Phox and Bem1p (PB1), a zinc finger domain, an LIR (LC3-interacting region) motif, and a ubiquitin-associated (UBA) domain [13]. P62 binds poly-ubiquitin chains via its C-terminal UBA domain and delivers ubiquitinated substrates to the autophagy system via its LIR domain and sends them to the proteasome via its PB1 domain [14,15]. Recently, NBR1 has also been identified as a cargo receptor and has a significantly similar structural domain organization to p62, including the PB1, ZZ, NBR1, and UBA structural domains, but NBR1 is twice the size of p62 and has other structural domains [16]. P62/NBR1 is involved in cell signaling and differentiation and plays the role of a signaling mediator downstream of the giant kinase titin [17,18]. 

Various roles have been reported for the NBR1 homologs in plants [19,20]. For instance, *AtNBR1* plays an important role in plant stress resistance because the *Arabidopsis nbr1* mutants present a senescence phenotype under different abiotic stress treatments [21], and *AtNBR1* acts as a selective autophagy receptor to mediate the selective autophagy of *AtExo70E2* [22]. NBR1 can bind to specific viral components or unknown bacterial proteins and mediate their elimination; it is beneficial to plant immunity in *Arabidopsis* [23,24]. Currently, in aggrephagy, AtNBR1 plays the role of plant receptor for cytosolic aggregates, and it is necessary to maintain proteostasis under heat and non-stress conditions [25]. *NtJoka2*, a tobacco NBR1 homolog, is located at the site of host–pathogen interactions and has shown an important role in plant resistance to fungal pathogens [20,26]. Moreover, NBR1 interacts with HSP90.1 and ROF1 and mediates their degradation by autophagy, which represses the response to heat stress by attenuating the expression of HSP genes regulated by the HSFA2 transcription factor [27]. However, the function of NBR1, which has a similar structure in wheat, is poorly understood in response to abiotic stresses.

In a previous study, we isolated several genes with unknown functions from the cDNA library by yeast two-hybrid screens; interestingly, there was an NBR1 homologous gene called *TaNBR1* [28]. In the current study, we explore the expression patterns of *TaNBR1* and its responses to different adverse treatments by gene expression analysis, and determine its subcellular localization. The transgenic *Arabidopsis* plants of overexpressing *TaNBR1* exhibited a significantly decreased drought tolerance and hypersensitive response to abscisic acid (ABA). Our results indicate that *TaNBR1* might play a negative role in drought stress tolerance in plants.

## 2. Results

### 2.1. The Molecular Characterization of TaNBR1

*TaNBR1* has an ORF of 2595 nucleotides, which encodes a protein containing 865 amino acids with a predicted molecular mass of 93.3 kDa and an isoelectric point of 5.91. The *TaNBR1* transcript sequence has been uploaded to NCBI (GenBank accession number MK419005). The *TaNBR1* gene sequence was compared to the values in the URGI database. The encoded sequence shares a high similarity (99%) with IWGSC RefSeq v1.0 chromosome 2B and the results obtained from the database show that *TaNBR1* is located on chromosome 2B in wheat. The genome sequencing results and cDNA sequence alignment show that the gene contains seven exons and six introns (Figure 1A).

Amino acid sequence analysis showed that TaNBR1 contains an N-terminal PB1 domain (amino acids 23-109), a zinc finger domain (amino acids 425-67), an NBR1 domain (amino acids 488-600), and a UBA domain (amino acids 798-835) at the C-terminal (Figure 1B). Amino acid sequence comparisons showed that the conserved domain above TaNBR1 protein shares a high sequence similarity with its homologs, i.e., a 62% similarity with *Oryza sativa* (XP_–_015625351); 60.8% with *Zea mays* (ACN33320); and 61.42% with TaUBA (ABB18390), but a low sequence similarity with its homologs *Vitis vinifera* (XP_–_002277480) and *Sorghum bicolor* (XP_–_002454107) (Figure 1C), thereby suggesting that the *TaNBR1* gene may be conserved in monocots. Evolutionary analysis of the full-length amino acid sequence showed that *TaNBR1* shares high identity with NBR1 homologous gene proteins, such as *Vitis vinifera* and *Arabidopsis thaliana* (Figure 1D). 

### 2.2. TaNBR1 Expression Profiles

Tissue expression analysis in wheat demonstrated the presence of *TaNBR1* in various organs, including the roots, leaves, spikelet, and stems, and the *TaNBR1* expression levels were highest in the leaves, in comparison to the other organs (Figure 2A). Furthermore, the expression levels of *TaNBR1* were very low in the roots and stems (Figure 2A). We analyzed the expression level of *TaNBR1* under different abiotic stresses. Under ABA treatment, the expression level of *TaNBR1* rapidly increased and reached its maximum at 3 h, before decreasing (Figure 2B). Undertreatment with NaCl, the mRNA expression level of *TaNBR1* gradually increased to peak at 6 h, after which it sharply decreased (Figure 2C). Under PEG stress, the transcription level rapidly increased, peaked at 3 h, and then sharply declined to a low level (Figure 2D). The *TaNBR1* transcript abundances changed very minimally under cold and heat stress (Figure 2E,F). These results demonstrate that *TaNBR1* responds rapidly to treatment with ABA and PEG, but more slowly to NaCl, and the sensitivity to cold and heat treatment is low. Therefore, we then focused on analyzing the function of *TaNBR1* in drought resistance.

### 2.3. TaNBR1 Localizes in the Golgi Apparatus and Autophagosomes

We fused the EGFP reporter gene construct to *TaNBR1* in pCAMBIA1302-EGFP, and the tobacco leaf cells transiently expressed the fusion protein vector 35S: TaNBR1-EGFP via an *Agrobacterium*-mediated transformation to observe the subcellular localization of TaNBR1. To further confirm the localization, we co-expressed TaNBR1-EGFP with several marker genes of distinct subcellular compartments in tobacco leaf cells. The subcellular location of TaNBR1-EGFP was generally identical to those of AtMemb12-RFP and AtATG8-RFP (Figure 3A–F). These results show that TaNBR1 localizes in the Golgi apparatus and autophagosomes.

### 2.4. The Characteristics of TaNBR1 Overexpressing Transgenic Arabidopsis

To determine the function of *TaNBR1*, the recombinant plasmid 35S: TaNBR1-EGFP was transferred into the *Agrobacterium tumefaciens* strain GV3101 and then transformed into *Arabidopsis*. Three homozygous transgenic T3 lines designated as OE-6, OE-10, and OE-15 were then investigated by Western blot analysis (Figure 4D). The 3 transgenic lines and WT control were cultured under the same conditions for 3 weeks; after not being watered for 2 weeks, most of the leaves wilted and withered in the *TaNBR1* transgenic lines, but the WT control lines continued to grow normally. Subsequently, after re-watering for 3 days, most of the transgenic lines did not survive, whereas the WT plants exhibited relatively good resilience and normal growth was restored (Figure 4A). Therefore, the survival rates of the transgenic plants were lower than those of the WT control (Figure 4C). In addition, the *TaNBR1* transgenic plants had relatively higher water loss rates and lower relative water contents, in comparison to the control according to the measurements of the relative water contents and water loss rates (Figure 4D,E). Therefore, these results suggest that the overexpression of *TaNBR1* can reduce drought resistance in *Arabidopsis*.

Under normal growth conditions, the chlorophyll, proline, malondialdehyde (MDA), superoxide dismutase (SOD), peroxidase (POD), and catalase (CAT) contents were not significantly different in the transgenic *TaNBR1* plants and WT (Figure 5A–F). However, under drought stress, the MDA contents were higher in the transgenic lines than the control (Figure 5A), whereas the chlorophyll and proline contents were lower in the transgenic lines than the control (Figure 5B,C). Moreover, the antioxidant enzyme (POD, CAT, and SOD) contents were lower in the transgenic lines than in the control (Figure 5D–F). Our results indicate that the activity levels of the antioxidant enzymes might be associated with the ability to withstand drought stress in the *TaNBR1* transgenic lines. Hence, *TaNBR1* may play a negative role in regulating plant drought resistance.

### 2.5. The Effects on Seed Germination and Seedling Growth in TaNBR1 Overexpression Transgenic Plants under Drought Stress

To study the abiotic stress response in the *TaNBR1* transgenic plants, seed germination and seedling growth were evaluated in the transgenic *TaNBR1* lines and WT plants under drought stress conditions. The *TaNBR1* transgenic *Arabidopsis* seeds were germinated on 1/2 MS medium and 1/2 MS medium with mannitol to further evaluate the stress tolerance responses. The seed germination rate and seedling growth of the *TaNBR1* transgenic plants and the WT were not significantly different under normal growth conditions. However, the seedlings of the *TaNBR1* transgenic plants exhibited more growth inhibition than the WT seedlings under treatment with 200 mM of mannitol (Figure 6A). The results indicate that the seed germination rates are lower in the *TaNBR1* transgenic lines than the WT control under treatment with 200 mM of mannitol (Figure 6C). Furthermore, the root lengths did not significantly differ in the *TaNBR1* transgenic plants and WT under normal growth conditions (Figure 6B–D). However, the primary roots of the *TaNBR1* transgenic plants were significantly shorter than those of the control under drought stress conditions (Figure 6B–D). These results indicate that the transgenic plants overexpressing *TaNBR1* are sensitive to drought stresses.

### 2.6. The Overexpression of TaNBR1 in Plants Sensitive to ABA

ABA signaling is frequently affected by abiotic stresses in plants [29]. We found that the transcription expression level of *TaNBR1* rapidly increased under ABA treatment, thereby suggesting that *TaNBR1* might be related to the ABA signaling pathway. We studied the responses of plants overexpressing *TaNBR1* under exogenous ABA treatment during seed germination and the post-germination growth phase to further investigate the involvement of ABA in the responses to abiotic stresses in transgenic plants. When the *TaNBR1* transgenic plants and WT were grown in 1/2 MS medium, seedling emergence did not significantly differ in the *TaNBR1* transgenic and WT plants. However, when grown in 1/2 MS medium with 2 μm of ABA, seedling emergence was inhibited in the transgenic plants, in comparison to the WT plants (Figure 7A). Furthermore, the seed germination rates were lower in the transgenic lines overexpressing *TaNBR1* than in the WT plants (Figure 7B). These results suggest that the seeds of the transgenic plants overexpressing *TaNBR1* are much more sensitive to ABA treatment than those of the WT in the seed germination stage. Thus, the transgenic *Arabidopsis* plants exhibited an ABA hypersensitivity phenotype during post-germination growth, and *TaNBR1* may be regulated via the ABA signaling pathway during plant responses to abiotic stresses.

### 2.7. The Effects of TaNBR1 on the Expression of Stress-Responsive Genes

To investigate the molecular mechanism involved in the regulation of *TaNBR1* under drought conditions, we conducted a quantitative expression analysis for six stress-associated genes in transgenic *Arabidopsis* and WT plants (Appendix A). Under normal growth conditions, the transcript expression levels of *P5CS1*, *RD29A*, *COR15A*, *RAB18*, *LEA14*, and *NCED3* did not exhibit obvious differences in the WT control and *TaNBR1*-overexpressing lines. However, under drought conditions, the expression of these genes was rapidly induced and to a much higher level, and the expression levels of these genes were lower in the *TaNBR1*-overexpressing seedlings than the WT controls (Figure 8). The results are consistent with the drought stress-hypersensitivity phenotypes determined in the *TaNBR1*-overexpressing lines grown in soil. Therefore, *TaNBR1* may regulate stress-associated genes to negatively regulate drought resistance in plants.

## 3. Discussion

Many studies have shown that the autophagy pathway plays an important role in abiotic stresses in plants [30,31]. However, little is known about the functions of NBR1 homolog genes in wheat. In the present study, we identified *TaNBR1* with the similar domain of NBR1 homologs and assessed its functions in drought tolerance in transgenic *Arabidopsis* to broaden our understanding of the functions of NBR1 genes in wheat.

In rice, the UBA protein OsDSK2a mediates seedling growth and salt responses, where it targets the brassinosteroid regulator BES1 to mediate selective autophagy during stress, thereby balancing plant growth and abiotic stress responses [32,33]. In addition, Arabidopsis UBAC2A and UBAC2B (UBA protein 2a/b) are partially redundant and they play critical roles in the heat tolerance of plants [34]. Thus, genes encoding the UBA domain are involved in a plant’s tolerance to abiotic stresses. In *Arabidopsis*, the NBR1 protein has multiple effects on regulating plant stress resistance, such as heat, oxidative, drought, salt, and necrotrophic pathogens [21]. In tomatos, both mutants and silenced strains of NBR1 exhibit increased the tolerance to heat stress by regulating the accumulation of ubiquitinated substrates. The transgenic lines, as well as autophagy defective mutants, cause obvious symptoms of wilting and drought stress and the seedlings are salt intolerant [21,35]. In addition, the knockout of *SmNBR1* shows an increased sensitivity to starvation and stress conditions and influences sexual structural development [36]. Thus, NBR1 homolog genes are involved in plant tolerance to positively regulate abiotic stresses. In our study, amino acid sequence analysis showed that TaNBR1 has higher similar conserved domains with the plant NBR1 homologs (Figure 1B,C). The quantitative results for the different tissues showed that *TaNBR1* gene expression was the highest in the leaves, but lowest in the roots (Figure 2A). Su et al. also found a similar phenomenon in *Populus*. They suggested that NBR1 resized the adversity by affecting the photosynthesis and ROS system [37]. In addition, the leaves were important sites for photosynthesis and the ROS system, so high expression levels were evident in the leaves. The quantitative results obtained after various stress treatments indicate that the expression of *TaNBR1* can be strongly induced by salt and drought stresses (Figure 2B–F). In drought stress, *TaNBR1* may act as a receptive gene, thus peaking at 3 h of stress and then transmitting the signal to downstream genes. Mao et al. also found a similar phenomenon that *TaSNAC13* peaked 6 hours after a drought and then sharply declined [38]. These results suggest that *TaNBR1* may be involved in abiotic stress responses in plants.

The plant trans-Golgi network is a multi-functional organelle that orchestrates the traffic of transport vesicles between vacuoles, the plasma membrane, late endosomes, and the Golgi apparatus. Both Rab11s and RABA1 are localized to the trans-Golgi network, where they mediate vesicle transport to the plasma membrane and/or PVC/LE, thereby indicating that there is a substantial link between trans-Golgi network trafficking and plant responses to stress [39,40]. In the present study, we found that TaNBR1 is localized in the Golgi apparatus and autophagosomes (Figure 3). Therefore, we consider that the regulatory effect of TaNBR1 on drought tolerance in plants may be related to trans-Golgi network trafficking.

Plants have evolved a range of mechanisms to deal with drought stress, for instance, increasing osmoregulatory substances and reducing transpiration rate. Therefore, various physiological, biochemical, and morphological parameters related to drought resistance can be used to evaluate the drought resistance of plants [41], such as the proline, chlorophyll, and MDA contents, and relative water content. The overexpression of *ZmPP2C-A2* in *Arabidopsis* enhances the water loss rate under drought stress to negatively regulate the response of plants to drought stress [42]. In addition, in comparison to WT plants, the loss of function by *OsASR5* leads to a greater sensitivity to drought stress and lower relative water contents under drought stress conditions [43]. Similar results were obtained in our study under drought stress conditions, where *TaNBR1*-overexpressing lines had lower relative water contents and higher water loss rates compared with WT plants (Figure 4E,F). MDA damages cellular activities and ion toxicity under stress conditions [44]. We found that the transgenic plants had higher MDA contents than the WT under drought conditions (Figure 5A), which indicated that the capacity to protect against cell damage was decreased. Similar results were previously obtained for plants overexpressing *GhWRKY6* [45]. To counter oxidative stress, plants possess an antioxidant system that continuously removes harmful reactive oxygen species. In our previous study, we showed that *Arabidopsis* plants overexpressing *TaDIS1* had lower POD, SOD, and CAT contents under drought stress conditions and they exhibited decreased drought tolerance [46]. Similarly, the transgenic plants tested in the present study had lower antioxidant enzyme levels than the WT (Figure 5D,F). Proline and chlorophyll are considered to affect their ability to adapt to stressful conditions. AtDi19-3 as a negative regulator, chlorophyll, and proline contents were lower in the seedlings overexpressing AtDi19-3 than in WT under drought stress [47]. In the present study, we found that the proline and chlorophyll contents were also lower in *TaNBR1* transgenic plants than the WT control under drought stress conditions (Figs 5B-C), thereby suggesting that the overexpression of *TaNBR1* decreased their ability to adapt to drought stress. Our results suggest that *TaNBR1* reduces the resistance of plants to drought by affecting the metabolic regulation of intracellular osmotic substances.

Previous studies have demonstrated that stress resistance in plants is related to germination and root growth by seedlings [46,48]. In the current study, the phytohormone ABA is involved in a wide range of biological processes, including seed maturation, regulating seed dormancy and germination, and plant growth and development under various abiotic stress conditions. Additionally, NBR1 is a regulator of ABA-dependent signaling pathways.We found that *TaNBR1*-overexpressing plants exhibited a significantly greater growth inhibition with shorter roots and they were sensitive to drought stress [49,50,51]. Recently, it was found that the NBR1 homologous gene AtNBR1 regulates the ABA signaling pathway by interacting with ABA pathway regulatory factors (ABI3, ABI4, and ABI5) to regulate seed germination, stomatal aperture, and root growth [52]. In our study, the overexpression of *TaNBR1* resulted in a greater sensitivity to ABA and drought stress, thereby suggesting that *TaNBR1* might be involved in the response to drought stress via ABA-related signaling pathways. 

It is widely accepted that plants defend themselves against drought stress by regulating the expression of stress-related genes, and ABA-dependent and ABA-independent pathways are involved in the drought stress response. Thus, to understand the molecular functional role of in drought stress responses, we employed qRT-PCR to analyze ABA-independent pathway-marker genes (*P5CS1* and *RD29A*) and ABA-dependent pathway-marker genes (*COR15A*, *RAB18*, *LEA14*, and *NCED3*). Under drought stress, previous studies showed that transgenic *Arabidopsis* plants overexpressing *TaNBR1* increased their drought tolerance by up-regulating these marker genes, compared with those in the WT [48,53,54]. In the present study, the overexpression of *TaNBR1* decreased the drought tolerance in *Arabidopsis* by down-regulating the expression of these marker genes, and thus *TaNBR1* may respond to drought stress by regulating these stress-related genes.

The homologous genes related to NBR1 in plants have mainly been reported in *Arabidopsis AtNBR1* and *Nicotiana NtJoka2*. Functional studies have shown that they play a conserved role in selective autophagy, because they have a similar domain organization and binding ability to ATG8 and ubiquitin [55]. 

In our research, we cloned a conserved domain similar to NBR1, but which negatively regulates drought conditions. The reasons may be as follows: the TaNBR1 we cloned is located in the Golgi apparatus and autophagosomes; AtNBR1 is not located in the Golgi apparatus but in the autophagosomes [22]. NBR1 can interact with ATG8 in yeast in both Arabidopsis and tobacco, but we found TaNBR1 in our research. There is no interaction with TaATG8 in yeast double hybrids, and we also confirmed that they do not interact in the BIFC experiment. In addition, BD-TaNBR1 alone has a significant self-activation effect, which is similar to NBR1 in Arabidopsis and tobacco. BD-TaNBR1 has a self-activation phenomenon, which is also completely different from Arabidopsis. In addition, previous studies have shown that the evolutionary analysis of NBR1 in different species has a relatively distant relationship, which also indicates that NBR1 has different functions. Interestingly, in another study, we also found that the overexpression of TaUBA, which also has a similar domain to NBR1, with the 35S promoter also negatively regulates drought stress responses in *Arabidopsis*. In addition, we also proved that TaUBA has the same localization as TaNBR1, and there is no interaction between it and TaATG8, which also shows that, although the TaNBR1 we cloned has a conserved domain similar to NBR1, its function is significantly different from NBR1.

In conclusion, we identified *TaNBR1* as having similar domains to NBR1 in wheat and characterized its biological function. Our results show that the expression level of *TaNBR1* is induced by drought stress and exogenous ABA treatment. *TaNBR1*-overexpressing lines exhibited an increased sensitivity to drought stress and exogenous ABA and reduced the resistance to drought stress. Therefore, our results demonstrate that *TaNBR1* negatively regulates the drought stress response via regulating stress-related genes. Our findings may be valuable for improving drought tolerance in wheat.

## 4. Materials and Methods

### 4.1. Plant Materials and Growth Conditions

The seeds of common hexaploid wheat (*Triticum aestivum* L. cv. Chinese Spring) were grown in Petri dishes in a growth environment with a 16 h light/8 h dark photoperiod at 26 °C, and under a light intensity of 10000 Lux. In the stress treatments, 2-week-old seedlings were treated with heat (42 °C), NaCl (200 mM), cold (4 °C), 0.2 mM of ABA, or 15% of PEG6000. Leaf tissues were sampled at 0, 3, 6, 9, 12, and 24 h after each different stress treatment. Plant materials were obtained from the root, stem, and leaf grown in a plate, and the panicle tissues for tissue-specific expression analysis. The *Arabidopsis* thaliana Columbia ecotype and overexpression plants were grown under long-day conditions (photo period = 16 h light/8 h dark) at a relative humidity of 50% and 22 °C. Tobacco (*Nicotiana*) plants were grown under the same conditions described above. One-month-old tobacco plants were used for *Agrobacterium* injection experiments.

### 4.2. The Isolation of TaNBR1 and Sequence Analysis

The *TaNBR1* gene was cloned using the total RNA obtained from common hexaploid wheat seedling leaf samples by reverse transcription PCR (RT-PCR). The primers TaNBR1-F1/R1 were designed to amplify the open reading frame (ORF) fragment of *TaNBR1*. The PCR product was obtained and constructed with the TA cloning vector to determine the full-length clone of *TaNBR1* by sequencing. To determine the gene structure of *TaNBR1*, we performed PCR amplification using the genomic DNA (gDNA) template. The leaves of Chinese Spring were used to extract DNA with the CTAB method [56]. We used 80 ng of genomic DNA as the template, and amplified the gene with the gene-specific primers gNBR1-1F/1R and gNBR1-2F/2R. The primers are listed in Appendix A.

Sequence assembly analyses were conducted using DNAMAN software. Primary structural analysis was performed using SMART (http://smart.embl-heidelberg.de, accessed on 20 March 2022). Multiple-protein alignment was performed with DNAMAN software. The phylogenetic tree was constructed using MEGA6 software (http://www.megasoftware.net/mega.html, accessed on 20 March 2022). The molecular size and isoelectric point of the protein were analyzed with the Compute pI/Mw tool via the ExPASy website.

### 4.3. RNA Isolation and Quantitative real-Time PCR (qRT-PCR)

The total RNA was extracted from various organs in the wheat and Arabidopsis materials subjected to different stress treatments with a TRIzol reagent (TaKaRa Biotechnology, Dalian, China). After drought stress for 10 days, the leaves were sampled from the Arabidopsis seedlings. After confirming that the concentration and quality of the measured RNA samples satisfied the requirements, cDNA was synthesized according to the manufacturer’s instructions (Invitrogen, Carlsbad, USA). The cDNA templates were then used for PCR amplification. qRT-PCR was conducted with the Light Cycler^®^ 96 detection system (Roche). The reaction procedure was as follows: denaturation at 95 °C for 10 min, followed by 46 cycles at 95 °C for 10 s, 58 °C for 10 s, 72 °C for 10 s, and 72 °C for 6 min. The PCR reaction system contained 12.5 μL of 2× Fast-Start Essential DNA Green Master (Roche, Germany), 120 ng of cDNA template, 1 μL of each primer (10 μM), and double-distilled H_2_O was added to make up the final volume to 25 μL. The actin genes in wheat and Arabidopsis served as the loading controls. The relative expression levels of the genes were analyzed by the 2^−∆∆Ct^ method [57]. The specific TaNBR1 quantitative primers and stress-related gene primers are listed in Appendix A.

### 4.4. The Subcellular Localization of TaNBR1t

The full-length gene coding region of *TaNBR1* was obtained from the TA cloning vector containing the *TaNBR1* gene fragment. The amplification product was then fused to the CaMV35S promoter and inserted upstream of the enhanced green fluorescence protein (EGFP) coding region in pCAMBIA1302-EGFP. The plasmid 35S: TaNBR1-EGFP was translated into *Agrobacterium* strain GV3101, and then injected with different Golgi apparatus-localized marker genes and into tobacco leaves, as previously described [58]. AtGos11 (trans-Golgi localized SNARE 11 protein) [59] and AtATG8 (autophagosomes) were obtained from the cDNA of *Arabidopsis thaliana* seedlings and constructed in the pCAMBIA1302-RFP vector to obtain p35S-Gos11-RFP and p35S-AtATG8-RFP, respectively. The constructed vectors were transferred into *Agrobacterium* strain GV3101, mixed with 35S: TaNBR1-EGFP, and then injected into 3–4-week-old tobacco leaves. The leaves were then examined at 40 h after infiltration using an Olympus confocal laser scanning microscope (LSCM IX83-FV1200).

### 4.5. The Genetic Trnasformation of Arabidopsis thaliana

The recombinant plasmid 35S: TaNBR1-EGFP constructed in the subcellular localization experiment was converted into *Agrobacterium* strain GV3101, and the transformation was then performed in *Arabidopsis* using the vacuum infiltration method [60]. The initial transgenic T0 lines were selected on 1/2 Murashige and Skoog (MS) medium with 50 mg/L of hygromycin, and then further detected by Western blot analysis with anti-GFP (MBL). The representative homozygous T3 generation seeds obtained were used in our study.

### 4.6. The Physiological Characteristics of the Transgenic Lines

The *TaNBR1* transgenic and wild-type (WT) plants were used to assess different stress tolerance responses. For the germination and root growth analyses, approximately 200 seeds from the WT *Arabidopsis* and *TaNBR1* transgenic plants were subjected to surface sterilization with 3% (v/v) of sodium hypochlorite for 10–12 min, before washing 5 times in sterile water. The seeds were planted on 1/2 MS medium with 200 mM of mannitol for the drought stress experiment and with 2 μM of ABA for the ABA-response experiment, before incubating at 4 °C for 3 days, and then moving to 22 °C. The seed germination rate was assessed each day for 2–8 days. To determine the survival rate, *Arabidopsis* seeds were germinated on 1/2 MS medium, transferred to soil, and grown for 3 weeks under conditions (photo period 16 h light/8 h dark) with 50% relative humidity at 22 °C. They were then subjected to drought treatment by not being watered for 2 weeks. The survival rate was measured after resuming watering for 3 days. To determine the water lost from the leaves, we obtained leaf materials from the transgenic and WT control *Arabidopsis* plants. The leaves were placed on filter paper and the moisture lost from the leaves was measured every 20 min, while incubating at a constant temperature of 25 °C, as described in a previous study [61]. After subjecting the WT and transgenic plants to drought treatment under the same conditions for 10 days, the leaves from the untreated control and drought-treated plants were employed to measure various physiological indices. First, 0.2 g of each sample was washed in a pre-cooled mortar, before adding 0.8 mL of 50 mol/L pre-cooled sodium phosphate buffer (pH 7.6). The mixture was homogenized, transferred to a centrifuge tube, and centrifuged at 12,000× *g* and 4 °C for 20–30 min. The supernatant was then used to measure the superoxide dismutase (SOD), peroxidase (POD), and catalase (CAT) activity levels according to the previously described methods [46]. Leaf samples from each line weighing 0.2 g were placed in 10 mL of 95% alcohol and centrifuged, before making up 5 mL of the supernatant to a volume of 25 mL with deionized water. The chlorophyll content was determined with a spectrophotometer and calculated according to a previously described method [62]. The proline and malondialdehyde (MDA) contents were measured, as previously described [63].

### 4.7. Statistical Analysis

All of the data were tested by analysis of variance using SPSS software (SPSS USA). Significantly different results were analyzed with Duncan’s test (*p* < 0.05 or *p* < 0.01). The figures were prepared with GraphPad Prism 7.

## Figures and Tables

**Figure 1 ijms-23-04519-f001:**
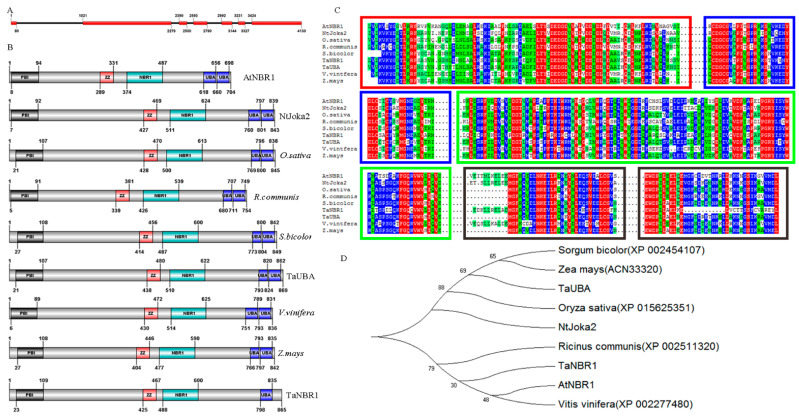
The *TaNBR1* gene structure, multiple alignments, and phylogenetic relationships in different species. (**A**) Genomic structure of *TaNBR1*, where the exons are indicated by red boxes and the introns by lines. The numbers represent the locations of the exons and introns. (**B**) The schematic diagram of the amino acid structure of NBR1 homologous genes in plants. (**C**) Multiple alignments of conserved domains in TaNBR1 and other related NBR1 homolog proteins in *Oryza sativa* (XP_–_015625351), *Aegilops tauschii* (XP_–_020187605.1), *Zea mays* (ACN33320), TaUBA (ABB18390), *Vitis vinifera* (XP_–_002277480), and *Sorgum bicolor* (XP_–_002454107). The alignment was performed using DNAMAN software. The conserved PB1 domain is marked with a red box, the ZnF–ZZ domain is indicated by a blue box, the NBR1 domain is shown by a green box, and the ubiquitin-associated (UBA) domain is shown by a black box. (**D**) The phylogenetic relationships of the full-length TaNBR1 and its closely related NBR1 in other plants. The cladogram was constructed using MEGA6.0 software.

**Figure 2 ijms-23-04519-f002:**
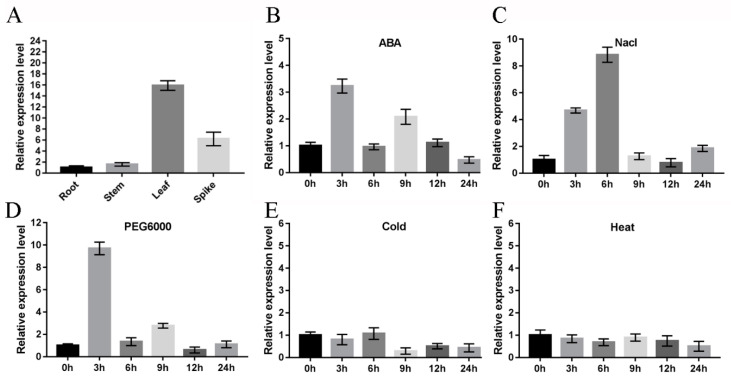
Expression analysis of *TaNBR1* in wheat. (**A**) Expression patterns of *TaNBR1* in different wheat tissues. (**B**–**F**) Expression patterns of *TaNBR1* under different stress treatments. Wheat seedlings were subjected to treatment with 0.2 mM abscisic acid (ABA), NaCl (200 mM), 15% PEG6000, cold (4 °C), and heat (42 °C), and the leaves were then collected at different time points, as indicated. *TaActin* was used as the internal control.

**Figure 3 ijms-23-04519-f003:**
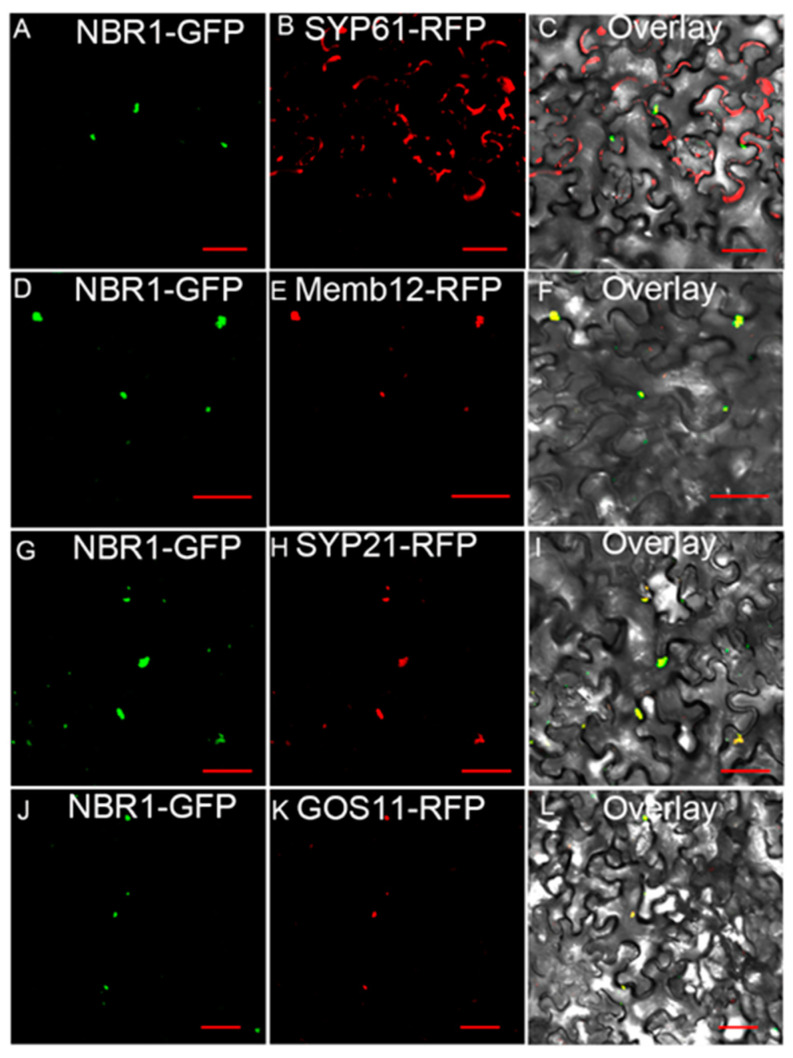
Subcellular localization of the TaNBR1 protein. 35S-NBR1-GFP and the indicated subcellular markers fused with fluorescent proteins were transformed into the epidermal cells of *Nicotiana benthamiana*. The single optical sections of the cells were imaged by confocal laser scanning microscopy. NBR1-GFP (**A**) is partially associated with the trans-Golgi network localized protein AtSYP61-RFP (**B**), as demonstrated in the overlay (**C**). NBR1-GFP (**D**) and AtMemb12-RFP (**E**) localized with the cis-Golgi localized marker, as demonstrated in the overlay (**F**). NBR1-GFP (**G**) co-localized with the prevacuolar compartment/late endosomes (PVC/LEs) marker AtSYP21-RFP (**H**), as demonstrated in the overlay (**I**). NBR1-GFP (**J**) co-localized with the trans-Golgi localized marker AtGos11-RFP (**K**), as demonstrated in the overlay (**L**). Bar = 50 μm.

**Figure 4 ijms-23-04519-f004:**
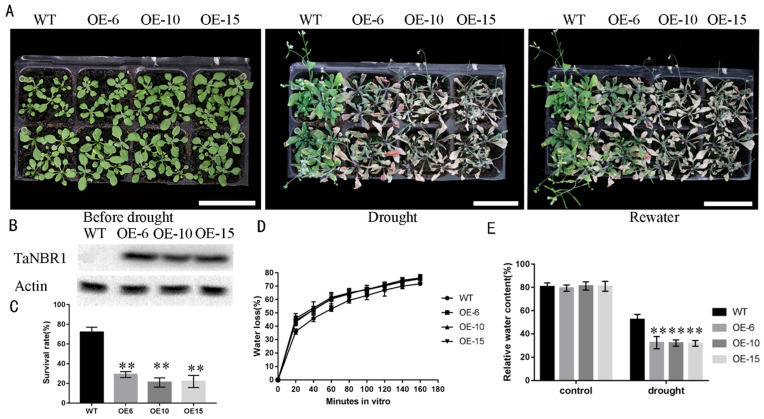
Phenotypes of the *TaNBR1*-overexpressing transgenic plants and wild-type controls in *Arabidopsis thaliana* under drought conditions. (**A**) Drought treatment using 3-week-old plants. The wild-type controls and transgenic plants were grown in pots for 3 weeks and then subjected to drought treatment for 2 weeks. The plants were watered again, 3 days after the treatment. Bar = 6 cm. (**B**) The protein levels determined in the transgenic plants. 35S: *TaNBR1*-EGFP was detected using an anti-GFP antibody and Actin was employed as the loading control detected with the anti-Actin antibody. (**C**) Survival rates of wild-type and *TaNBR1*-overexpressing transgenic *Arabidopsis* plants after resuming watering. (**D**) Water loss by *TaNBR1* overexpression transgenic and wild-type *Arabidopsis* plants in response to dehydration. (**E**) Relative water content in transgenic *Arabidopsis* plants and wild-type controls under normal conditions and drought stress. The data represent the mean ± standard deviation (n = 3), ** *p* ≤ 0.01.

**Figure 5 ijms-23-04519-f005:**
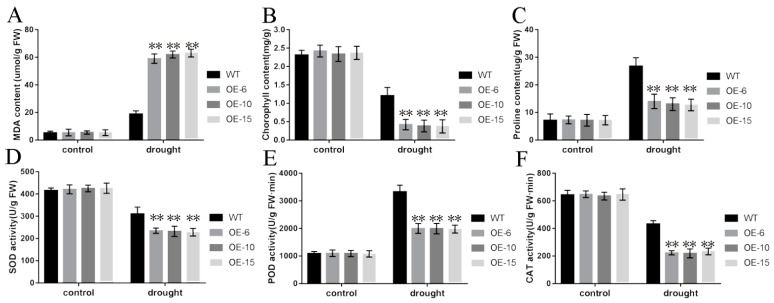
Physiological indexes determined in *TaNBR1*-overexpressing *Arabidopsis* transgenic plants and wild-type controls after drought treatment. (**A**) Malondialdehyde content. (**B**) Chlorophyll content. (**C**) Proline content. (**D**) Superoxide dismutase (SOD) content. (**E**) Peroxidase (POD) activity levels. (**F**) Catalase (CAT) activity levels measured before and after drought stress. Each data point is the average based on at least three replicates, ** *t*-test, *p* < 0.01.

**Figure 6 ijms-23-04519-f006:**
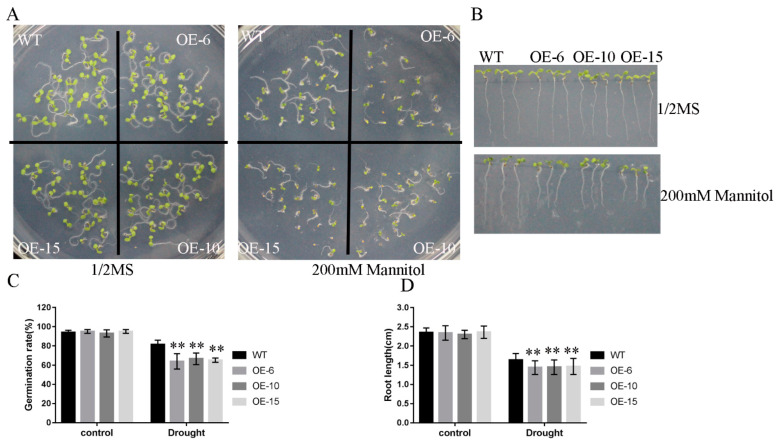
Drought stress responses of *TaNBR1*-overexpressing transgenic *Arabidopsis* and wild-type plants. (**A**) Seedling growth by wild-type and *TaNBR1*-overexpressing transgenic *Arabidopsis* plants under treatment with 200 mM of mannitol for 5 days. (**B**) Root length under the mannitol treatment. Surface-sterilized wild-type and transgenic *Arabidopsis* plant seeds were sown on 1/2 MS solid medium and incubated at 4 °C for 3 days. The seeds were transferred from 1/2 MS medium to plates containing 1/2 MS and 1/2 MS with 200 mM of mannitol, and the images were acquired after 5 days. (**C**,**D**) Seed germination percentage and root length in wild-type and transgenic *Arabidopsis* plants. The data represent the mean ± standard deviation (n = 3), and the asterisks over the bars indicate the significant differences between the wild-type and transgenic *Arabidopsis* lines. ** *t*-test, *p*< 0.01.

**Figure 7 ijms-23-04519-f007:**
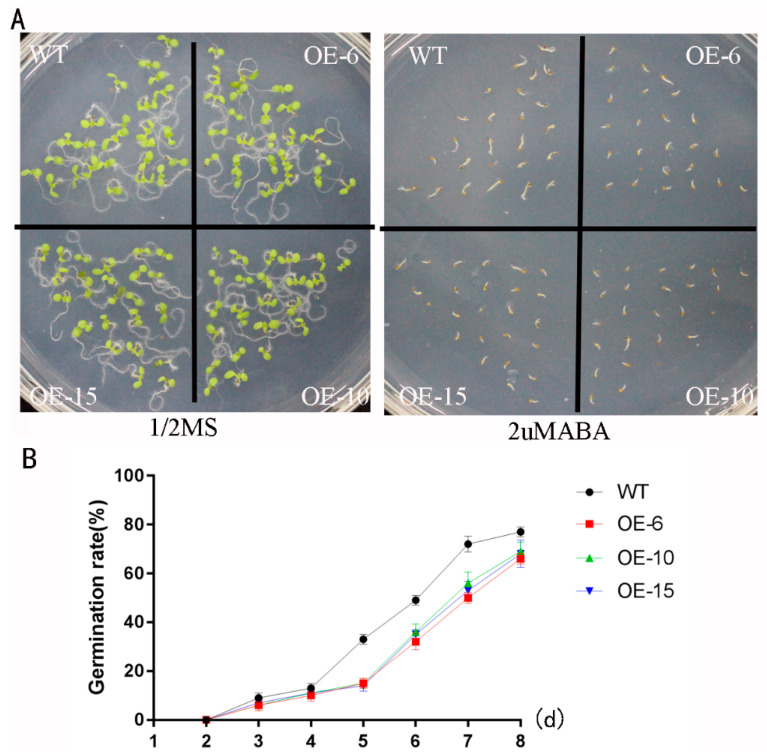
Phenotypes of wild-type and *TaNBR1*-overexpressing transgenic *Arabidopsis* lines under treatment with abscisic acid (ABA). (**A**) Seedling growth by wild-type and *TaNBR1*-overexpressing transgenic *Arabidopsis* lines on MS medium supplemented with or without 2 μM of ABA after incubation for 8 days at 4 °C for 3 days. (**B**) Seed germination percentage rates in wild-type and *TaNBR1*-overexpressing transgenic *Arabidopsis* lines. The data indicate the mean ± standard deviation (n = 3). Three independent replicates were conducted.

**Figure 8 ijms-23-04519-f008:**
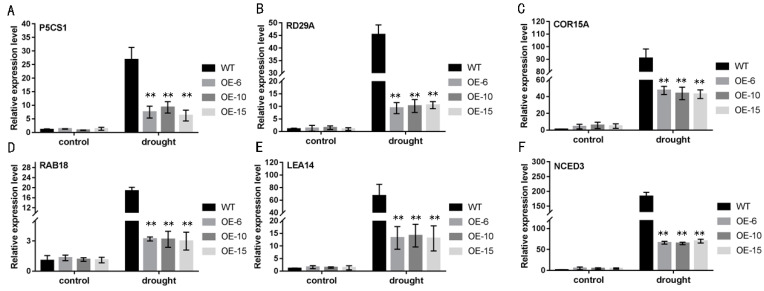
Expression levels of stress-related genes in wild-type and transgenic plants under drought stress conditions. The plants were grown in soil for 3 weeks and then not watered for 7 days, before determining the expression levels of P5CS1 (**A**), RD29A (**B**), COR15A (**C**), RAB18 (**D**), LEA14 (**E**), and NCED3 (**F**) by qRT-PCR. The actin gene was used as an internal reference. The asterisks over the bars indicate the significant differences between the wild-type and transgenic Arabidopsis lines. ** *t*-test, *p* < 0.01.

## Data Availability

The data presented in this study are available in the Appendix A.

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
