# Peer review of "TaNBR1, a Novel Wheat NBR1-like Domain Gene Negatively Regulates Drought Stress Tolerance in Transgenic Arabidopsis"

_ijms, 2022, doi:10.3390/ijms23094519_

Round 1
Reviewer 1 Report
The authors reported TaNBR1, a novel wheat NBR1 like domains gene, negatively regulates drought stress tolerance in transgenic Arabidopsis. This work may be acceptable for publication after addressing the comments below and revising the manuscript.
- The authors need to improve the introduction section (e.g., emphasis).
- Please add the comparison table among other literatures for the stress-related genes.
- The authors need to improve the discussion section.
Author Response
Response to Reviewer 1 Comments Point 1: The authors need to improve the introduction section (e.g., emphasis) Response 1: Thanks for your suggestion. We have improved the introduction section. And our manuscript has been corrected by some native–standard English speakers. Point 2: Please add the comparison table among other literatures for the stress-related genes. Response 2: We gratefully appreciate for your valuable suggestion. We had added Table.S2 about related genes. Point 3: The authors need to improve the discussion section. Response 3: Thanks for your instructive suggestion. Has been modified. We added the discussion about UBA protein domain and TaNBR1 subcellular localization.And condensed some sentences. And our manuscript has been corrected by some native–standard English speakers.Reviewer 2 Report
The manuscript (TaNBR1, a novel wheat NBR1 like domains gene, negatively regulates drought stress tolerance in transgenic Arabidopsis).
Please rephrase the following sentence to be understandable (In the ALP, a selective autophagy substrate is recognized and ubiquitinated by autophagy receptors such as p62/SQSTM1 and NBR1, which recruits substrate to the autophagosome for degradation by interacting with the LC3/ATG8 protein).
L93 Make TaNBR1 italic
L113 The following statement is incorrect (TaNBR1 increased gradually to peak at 6 h, before decreasing gradually). I can see that expression levels were sharply decreased after 6 h, please correct.
L170 Please write the full name of the enzymes (MDA, SOD, POD, and CAT) at the first time in the manuscript.
I found a very big difference in the expression between the roots and the leaves could you please explain?
In the discussion, could you please explain why the expression decreased quickly and sharply over time?
Please refer to the results of enzymes and gene expression in the conclusion.
Author Response
Response to Reviewer 2 Comments
Point 1: Please rephrase the following sentence to be understandable (In the ALP, a selective autophagy substrate is recognized and ubiquitinated by autophagy receptors such as p62/SQSTM1 and NBR1, which recruits substrate to the autophagosome for degradation by interacting with the LC3/ATG8 protein).
Response 1: Thanks for your suggestion. We have rewritten this sentence. (lines 39-42)
As the selective autophagy substrate p62/SQSTM1 and NBR1can be recognized and ubiquitinated by autophagy receptors in the ALP. In order to degrade, they recruit substrate to the autophagosome by interacting with the LC3/ATG8 protein.
Point 2: L93 Make TaNBR1 italic
Response 2: Thanks for your suggestion. We have made TaNBR1 italic. (line 98)
Point 3: L113 The following statement is incorrect (TaNBR1 increased gradually to peak at 6 h, before decreasing gradually). I can see that expression levels were sharply decreased after 6 h, please correct.
Response 3: Thanks for your suggestion. We have made correction according to the reviewer’s comments. (lines 119-120)
Point 4: L170 Please write the full name of the enzymes (MDA, SOD, POD, and CAT) at the first time in the manuscript.
Response 2: Thanks for your suggestion. We have improved it(lines 176-177)
Point 5: I found a very big difference in the expression between the roots and the leaves could you please explain?
Response 2: NBR1 is a candidate gene for selective autophagy receptors. Su et al. (2021) also showed that the NBR1 gene has different expressions in different tissues, with higher expression in leaves and xylem but a lower expression in roots. The main reason may be that NBR1 plays an important role in plant resistance and senescence. In addition, the ROS system regulates the adaptation to stress and senescence. Leaves are important organs for plant growth, so they have a high expression level in leaves.
- Su, W.; Lu, Y.; He, F.; Wang, S.; Wang, S., Poplar autophagy receptor nbr1 enhances salt stress tolerance by regulating selective autophagy and antioxidant system. Plant Sci. 2021, 11, 568-411
Point 6: In the discussion, could you please explain why the expression decreased quickly and sharply over time?
Please refer to the results of enzymes and gene expression in the conclusion.
Response 2: In drought stress, NBR1 may act as a receptive gene, thus peaking at 6h of stress and then transmitting the signal to downstream genes. Mao et al. also found a similar phenomenon that TaSNAC13 peaked 6 hours after drought and then declined sharply.
- Mao, H.; Li, S.; Wang, Z.; Cheng, X.; Li, F.; Mei, F.; Chen, N.; Kang, Z., Regulatory changes in TaSNAC8‐6A are associated with drought tolerance in wheat seedlings. Plant Biotechnol. J. 2020, 18, 1078-1092.